# The newly-arisen Devil facial tumour disease 2 (DFT2) reveals a mechanism for the emergence of a contagious cancer

Alison Caldwell[1,2], Rachel Coleby[1,2], Cesar Tovar[3], Maximilian R Stammnitz[4], Young Mi Kwon[4], Rachel S Owen[1], Marios Tringides[1], Elizabeth P Murchison[4], Karsten Skjødt[5], Gareth J Thomas[2,6], Jim Kaufman[4,7], Tim Elliott[2,6], Gregory M Woods[3], Hannah VT Siddle[1,2]*

[1]Department of Biological Sciences, University of Southampton, Southampton, United Kingdom; [2]Institute for Life Sciences, University of Southampton, Southampton, United Kingdom; [3]Menzies Institute for Medical Research, University of Tasmania, Hobart, Australia; [4]Department of Veterinary Medicine, University of Cambridge, Cambridge, United Kingdom; [5]Department of Cancer and Inflammation, University of Southern Denmark, Odense, Denmark; [6]Centre for Cancer Immunology, Faculty of Medicine, University of Southampton, Southampton, United Kingdom; [7]Department of Pathology, University of Cambridge, Cambridge, United Kingdom

*For correspondence:
H.V.Siddle@soton.ac.uk

Competing interests: The authors declare that no competing interests exist.

**Abstract** Devil Facial Tumour 2 (DFT2) is a recently discovered contagious cancer circulating in the Tasmanian devil (*Sarcophilus harrisii*), a species which already harbours a more widespread contagious cancer, Devil Facial Tumour 1 (DFT1). Here we show that in contrast to DFT1, DFT2 cells express major histocompatibility complex (MHC) class I molecules, demonstrating that loss of MHC is not necessary for the emergence of a contagious cancer. However, the most highly expressed MHC class I alleles in DFT2 cells are common among host devils or non-polymorphic, reducing immunogenicity in a population sharing these alleles. In parallel, MHC class I loss is emerging in vivo, thus DFT2 may be mimicking the evolutionary trajectory of DFT1. Based on these results we propose that contagious cancers may exploit partial histocompatibility between the tumour and host, but that loss of allogeneic antigens could facilitate widespread transmission of DFT2.
DOI: https://doi.org/10.7554/eLife.35314.001

## Introduction

Contagious cancers have emerged and circulate in two species of mammals (dogs and Tasmanian devils) and four species of molluscs (*Metzger et al., 2015*, *2016*; *Novinski, 1876*; *Pearse and Swift, 2006*). The Tasmanian devil is the only mammalian species in which two independent contagious cancers exist, Devil Facial Tumour 1 (DFT1) and Devil Facial Tumour 2 (DFT2) (*Pye et al., 2016b*). Although both cancers have similar gross morphology, causing tumours on the face, neck and oral cavity, genetic analysis shows that they emerged in different individuals (*Pearse and Swift, 2006*; *Pye et al., 2016b*; *Stammnitz et al., 2018*). DFT1 was first identified in 1996 in the northeast of Tasmania (*Hawkins et al., 2006*), but has since spread widely, causing close to 100% mortality and drastic decline of affected populations (*Hamede et al., 2015*). In contrast, DFT2 is a more recent contagious cancer, it was first characterised in 2014 and has been formally identified in eleven devils from the Channel region of Southwest Tasmania (*Pye et al., 2016b*; *Kwon et al., 2018*). Recent immunohistochemical and drug sensitivity analyses of DFT2 indicate it arose from a similar tissue to DFT1 (*Stammnitz et al., 2018*).

**eLife digest** While cancer cells typically cannot spread between individuals, there are a few examples of contagious tumours. Remarkably, two examples have emerged in a single species, the Tasmanian devil, a marsupial carnivore. These tumours are known as Devil Facial Tumour 1 (or DFT1) and Devil Facial Tumour 2 (or DFT2); both cause tumours round the faces of infected devils. Since it emerged in the 1990s, DFT1 has killed 60–90% of the devil population. Preserving the few remaining healthy devils has been a major challenge for conservationists. This challenge became more urgent in 2014, when the second contagious cancer, DFT2, was discovered circulating in the population.

Contagious cancers are rare because the immune system usually eliminates cells coming from outside the body. Each healthy cell carries molecules known as the major histocompatibility complex (MHC) that act as a barcode showing where each cell comes from. The immune system uses these molecules to help it tell the difference between the body's own cells and those from elsewhere. DFT1 can avoid the immune system and spread to new hosts because it has lost MHC. In 2014, scientists identified a new Tasmanian devil cancer named DFT2, but it was unclear how this second cancer evades the immune system as it spreads from host to host.

Caldwell et al. have now examined the MHC on cells from DFT2 cancers, including cells grown in the laboratory and cells taken from cancer biopsies. Biochemical tests showed that the DFT2 cells do carry MHCs, but that the MHC barcodes of DFT2 are similar to those of the devils infected with the disease. This finding may explain how the cancer can spread undetected in these animals, because the immune system does not recognize it as coming from outside the body. Further analyses also reveal that the cancer cells are slowly evolving to lose their MHCs. This means DFT2 could, with time, become as contagious as DFT1.

These two contagious cancers threaten the future of the Tasmanian devil. As top predators, devils are key to the ecosystem in Tasmania and their preservation is vital. While DFT2 provides a unique opportunity to study an emerging cancer as it develops, this research will also help to protect the devils and may lead to effective vaccines. These results could also reveal how other cancers avoid the immune system and may help to detect them during treatment. In addition, there are many similarities between contagious cancer cells and organ transplants. Understanding the role of MHC in DFT2 could lead to better ways to prevent rejection following transplants.

DOI: https://doi.org/10.7554/eLife.35314.002

As allografts, DFT1 and DFT2 cells should be rejected by the host devil through T cell recognition of non-self MHC class I molecules encoded by the major histocompatibility complex (MHC) (*Sherman and Chattopadhyay, 1993*). The mature MHC class I molecule is a trimer that acts as a ligand for the T cell receptor (TCR) and is composed of a light chain, $\beta_2$-microglobulin ($\beta_2$m), in non-covalent association with a heavy chain, which provides a binding cleft for self and non-self peptides. Classical MHC class I molecules are important transplantation antigens that are highly polymorphic, ubiquitously expressed and involved in peptide-antigen presentation. In contrast, non-classical MHC class I molecules have little polymorphism and more diverse functions, including regulation of the immune response particularly by inhibiting natural killer (NK) cell function (*Braud et al., 1999*). Allografts can be rejected within 7–14 days when host CD8+ T cells are exposed to non-self MHC class I molecules and their bound peptides on the surface of donor cells (*Rosenberg, 1993*).

Vertebrates have multiple MHC class I genes that are generally not orthologous between species due to rapid evolution (*Adams and Parham, 2001*). Five MHC class I genes have been identified in the genomic region encoding the Tasmanian devil MHC (*Cheng et al., 2012*). *Saha-UA, -UB* and –*UC*, are likely classical, with moderate levels of polymorphism and ubiquitous expression, while *Saha-UD, –UK* and -*UM*, are non-classical, with limited polymorphism and restricted expression (*Cheng et al., 2012*; *Cheng and Belov, 2014*).

We have previously demonstrated that DFT1 cells down-regulate MHC class I and class II molecules from their cell surface, thereby removing the targets for a T cell response (*Siddle et al., 2013*). This is the first indication that DFT1 cells have similar immune evasion mechanisms to those found in single organism tumours (*Hicklin et al., 1999*; *McGranahan et al., 2017*). Loss of MHC class I is due

to epigenetic down-regulation of $\beta_2$m, and the peptide transporter encoded by TAP1 and TAP2 (*Siddle et al., 2013*) but MHC class I can be restored in DFT1 cells when treated with IFN$\gamma$ (*Siddle et al., 2013*). MHC class I positive DFT1 cells have since been used with some success in initiating immune responses to DFT1 cells (*Tovar et al., 2017*). Down-regulation of MHC is also observed in the canine transmissible venereal tumour (CTVT), where MHC molecules are down-regulated during transmission and growth (*Hsiao et al., 2002*; *Pérez et al., 1998*). However, subsequent cytokine signalling leads to lymphocyte infiltration of tumours, MHC up-regulation and either tumour regression or stasis of growth (*Yang et al., 1987*). These findings have suggested that loss of MHC class I expression is necessary for the emergence and subsequent transmission of contagious cancers.

Why DFT2 cells can pass between individuals without a protective immune response from the host is not known, nor are the conditions that have facilitated the emergence of two contagious cancers in this species. Here we show that in contrast to DFT1, DFT2 cells express MHC class I molecules. However, the dominantly expressed MHC class I alleles are either non-classical or shared with host devils and common in the wider population. Finally, DFT2 tumours were identified with evidence of MHC class I loss, indicating the emergence of MHC class I down-regulation in this tumour.

## Results

### DFT2 cells express $\beta_2$m protein, a component of the MHC class I complex

Flow cytometry using a monoclonal antibody against Tasmanian devil $\beta_2$m was used to assess the level of MHC class I expression on the surface of a representative DFT1 cell line (DFT1_4906) (*Figure 1A*), DFT1 cells treated with IFN$\gamma$ (DFT1_4906 + IFN$\gamma$) (*Figure 1B*), three DFT2 cell lines (DFT2_RV, DFT2_SN and DFT2_TD549) (*Figure 1C,D and E*) and devil fibroblast cells (Fibroblasts_Salem) (*Figure 1F*). As has been previously reported (*Siddle et al., 2013*), DFT1 cells do not express $\beta_2$m, but DFT1 cells treated with IFN$\gamma$ upregulate $\beta_2$m to the same level as devil fibroblast cells. In contrast, all three DFT2 cell lines express cell surface $\beta_2$m, but the level of expression varies, DFT2_TD549 cells have the highest level of expression (Mean Fluorescence Intensity (MFI) 42.1)

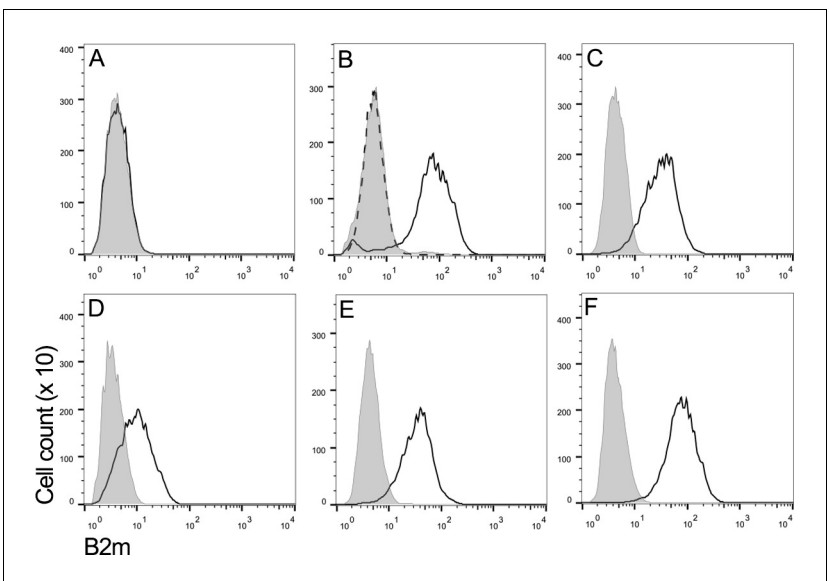

**Figure 1.** DFT2 cells express $\beta_2$m in vitro. Flow cytometry to compare $\beta_2$m expression shows (**A**) DFT1_4906, (**B**) DFT1_4906 + IFN$\gamma$, (**C**) DFT2_RV, (**D**) DFT2_SN, (**E**) DFT2_TD549 and (**F**) Fibroblast_Salem cells stained with $\alpha$-$\beta_2$m (solid line) and secondary only control (shaded peak). DFT1_4906 + IFN$\gamma$ (**B**) stained with blocked $\alpha$-$\beta_2$m antibody is also shown (dashed line). Fluorescence intensity for $\alpha$-$\beta_2$m on x-axis and cell counts on y-axis.
DOI: https://doi.org/10.7554/eLife.35314.003

followed by DFT2_RV (MFI 37.6) and DFT2_SN (MFI 12.1). Although all three DFT2 cell lines express $\beta_2$m, they have lower levels of $\beta_2$m than DFT1 cells that have been treated with IFN$\gamma$ (MFI 92.5) and devil fibroblasts (MFI 93.3). These results indicate that DFT2 cells grown in culture express surface MHC class I molecules in contrast to DFT1, which is MHC class I negative.

## DFT2 cells express classical and non-classical MHC class I heavy chain genes in vitro and in vivo

RT-qPCR was used to quantify the relative level of MHC class I gene expression for $\beta_2$m, classical MHC class I and non-classical MHC class I genes in vitro (*Figure 2A–D*). Due to high nucleotide similarity between the classical MHC class I genes in the Tasmanian devil, *Saha-UA, -UB* and *–UC*, and their respective alleles, a single primer set was used to amplify transcripts from these genes as a group. Gene specific primers were used to amplify the non-classical MHC class I genes, *Saha-UK* (*Figure 2C*) and *Saha-UD*, which was found only at trace levels (*Figure 2—figure supplement 1*). RT-qPCR shows that DFT2 cells express significantly higher levels of $\beta_2$m (*Figure 2A*), classical MHC

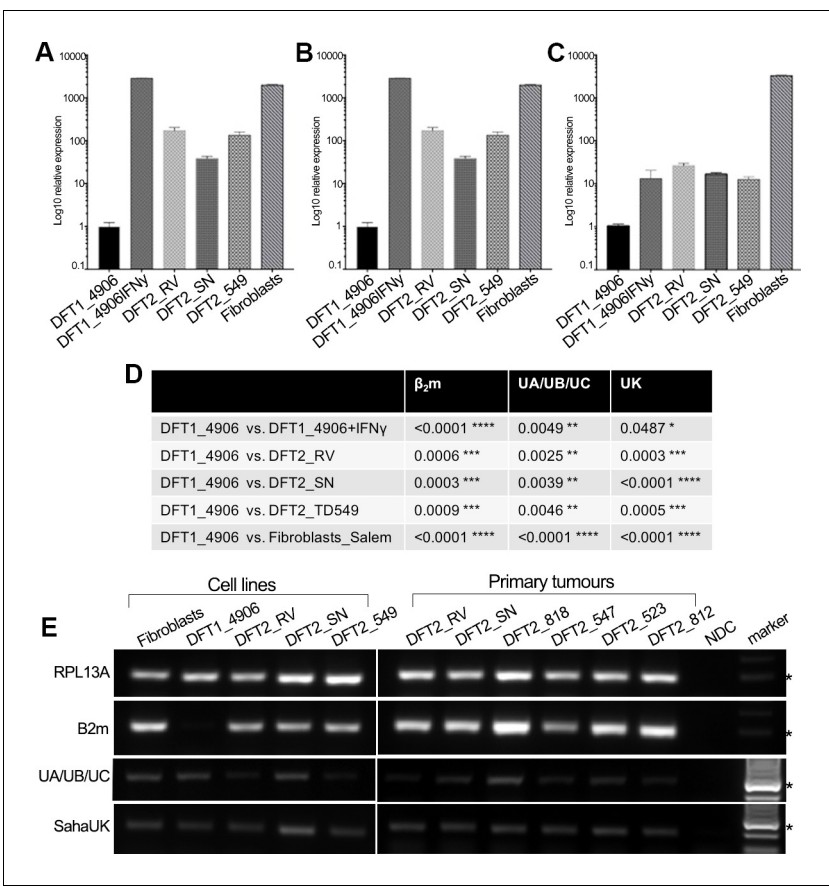

**Figure 2.** DFT2 cells in vitro and in vivo express mRNA for $\beta_2$m, *Saha-UK, Saha-UA, UB* and *UC*. RT-qPCR of (A) $\beta_2$m, (B) *Saha-UA, -UB* and *-UC* and (C) *Saha-UK* mRNA expression by DFT2 cell lines (DFT2_RV, DFT2_SN, DFT2_TD549), fibroblast cells (Fibroblasts_Salem) and DFT1 cells treated with IFN$\gamma$ (DFT1_4906 + IFN$\gamma$) relative to DFT1_4906 cells. Gene expression levels are normalized against RPL13A as a housekeeping gene. Data are represented as mean ± S.E.M of three technical replicates. (D) An unpaired T-test was performed to test for statistical significance. (E) RT-PCR on DFT2 cell lines and DFT2 primary tumours for $\beta_2$m, *Saha-UA, -UB* and *-UC* and *Saha-UK*. RPL13A was used as a loading control. A no DNA control (NDC) is included for each RT-PCR. A marker at 300 base pairs is indicated by an asterisk.

DOI: https://doi.org/10.7554/eLife.35314.004

The following figure supplement is available for figure 2:

**Figure supplement 1.** RT-PCR amplification of RPL13A and Saha-UD from DFT1, DFT2 and Fibroblast cells.
DOI: https://doi.org/10.7554/eLife.35314.005

class I heavy chain genes (*Saha-UA, -UB* and *-UC*) (*Figure 2B*) and non-classical MHC class I (*Saha-UK*) (*Figure 2C*) than DFT1_4906. However, expression of $\beta_2$m and *Saha-UA, -UB* and *-UC* by DFT2 cells is lower than that of DFT1_4906 + IFN$\gamma$ and Fibroblasts, which is consistent with the levels of $\beta_2$m expression observed on the DFT2 cell lines. Interestingly, while the levels of *Saha-UA, -UB* and *-UC* in the three DFT2 cell lines is lower than DFT1_4906 + IFN$\gamma$ (*Figure 2B and D*), the levels of *Saha-UK* are not significantly different (*Figure 2C and D*). This is despite the fact that the expression level of the classical MHC class I genes reflects the amplification of three different MHC class I loci compared to a single locus, *Saha-UK*. Thus, DFT2 cells in vitro express both classical and non-classical MHC class I heavy chain genes.

To extend this analysis to primary DFT2 tumours we used RT-PCR to amplify $\beta_2$m and MHC class I heavy chain transcripts from the DFT2 cell lines and six primary DFT2 tumours collected between March 2014 and January 2016 (*Figure 2E* and *Supplementary file 1*). The primary tumours include DFT2_RVT1 and DFT2_SNT2 from which the cell lines DFT2_RV and DFT2_SN were derived. RT-PCR shows evidence of expression for $\beta_2$m in the cell lines and in primary tumour samples (*Figure 2E*), reflecting the expression of cell surface $\beta_2$m protein in the cell lines. *Saha-UK* is expressed in all cell lines and primary biopsies. The cell lines and primary tumours express classical MHC class I, but the expression levels appear to be variable between the primary tumours. While this analysis is not quantitative, as the amount of stroma in each sample varies between tumours, these results show that DFT2 cells express both classical and non-classical MHC class I transcripts alongside $\beta_2$m.

## The expression of MHC class I molecules varies in DFT2 tumours in vivo

To further investigate the expression of MHC class I molecules between DFT2 tumours in vivo, a shared peptide immunogen was used to raise a pan-classical MHC I antibody against the classical MHC class I heavy chains (Saha-UA, -UB and -UC). A second peptide, specific in sequence to Saha-UK, was used to raise an antibody against the non-classical MHC class I, Saha-UK. Monoclonal antibodies were initially screened by western blot using protein from devil fibroblast cells. Positive clones were re-screened for molecule specificity against recombinant Saha-UK and recombinant Saha-UC protein (*Figure 3—figure supplement 1*). Clones specific for Saha-UK (clone - $\alpha$-UK_15-29-1) and Saha-UA –UB and -UC (clone - $\alpha$-UA/UB/UC_15-25-18) were identified.

Staining of DFT2 serial sections from six primary DFT2 tumours (*Supplementary file 1*) with these antibodies demonstrates expression of both classical (Saha-UA, -UB and –UC) and non-classical (Saha-UK) MHC class I molecules in vivo (*Figure 3* and *Figure 3—figure supplement 2*). However, this analysis also demonstrates that MHC class I expression is variable in DFT2 tumours. Three of the tumours, DFT2_RVT1, DFT2_SNT2 and DFT2_818T1 (*Figure 3*), retain strong expression of classical class I molecules, with localisation to the cell membrane. This result is consistent with the cell surface expression of $\beta_2$m observed on the DFT2_RV and DFT2_SN cell lines, derived from two of these primary tumours (*Figure 1C and D*). However, expression of classical MHC class I in DFT2_547 and DFT2_523 is weaker, appears mostly cytoplasmic and shows some variation in staining intensity, with some cells in DFT2_547 showing very low levels of expression. Strikingly, DFT2_812 is negative for classical MHC class I (*Figure 3*).

The expression of Saha-UK also varies between and within DFT2 biopsies (*Figure 3*). DFT2_RVT1 cells are uniformly positive for Saha-UK, whereas DFT2_SNT2 and DFT2_547 have variable expression, with some areas staining more weakly. Interestingly, there is some evidence for Saha-UK staining in DFT2_812, a tumour negative for classical MHC class I, but DFT2_523 is largely Saha-UK negative (*Figure 3*). While the DFT2 tumours with strong classical class I expression show localisation to the cell membrane (i.e. cell surface expression), staining for Saha-UK shows a cytoplasmic distribution in some of the DFT2 tumours.

Notably, DFT2 tumours also show morphological heterogeneity (*Figure 3—figure supplement 3*). DFT2 tumour cells are sometimes arranged as nests separated by a fibroblastic stroma, for example DFT2_547T1, but can also be arranged in diffuse sheets separated by a fibrous component, for example DFT2_818T1 and DFT2_523T1. The degree of individual tumour cell nuclear pleomorphism can also vary within the same tumour (DFT2_SNT2).

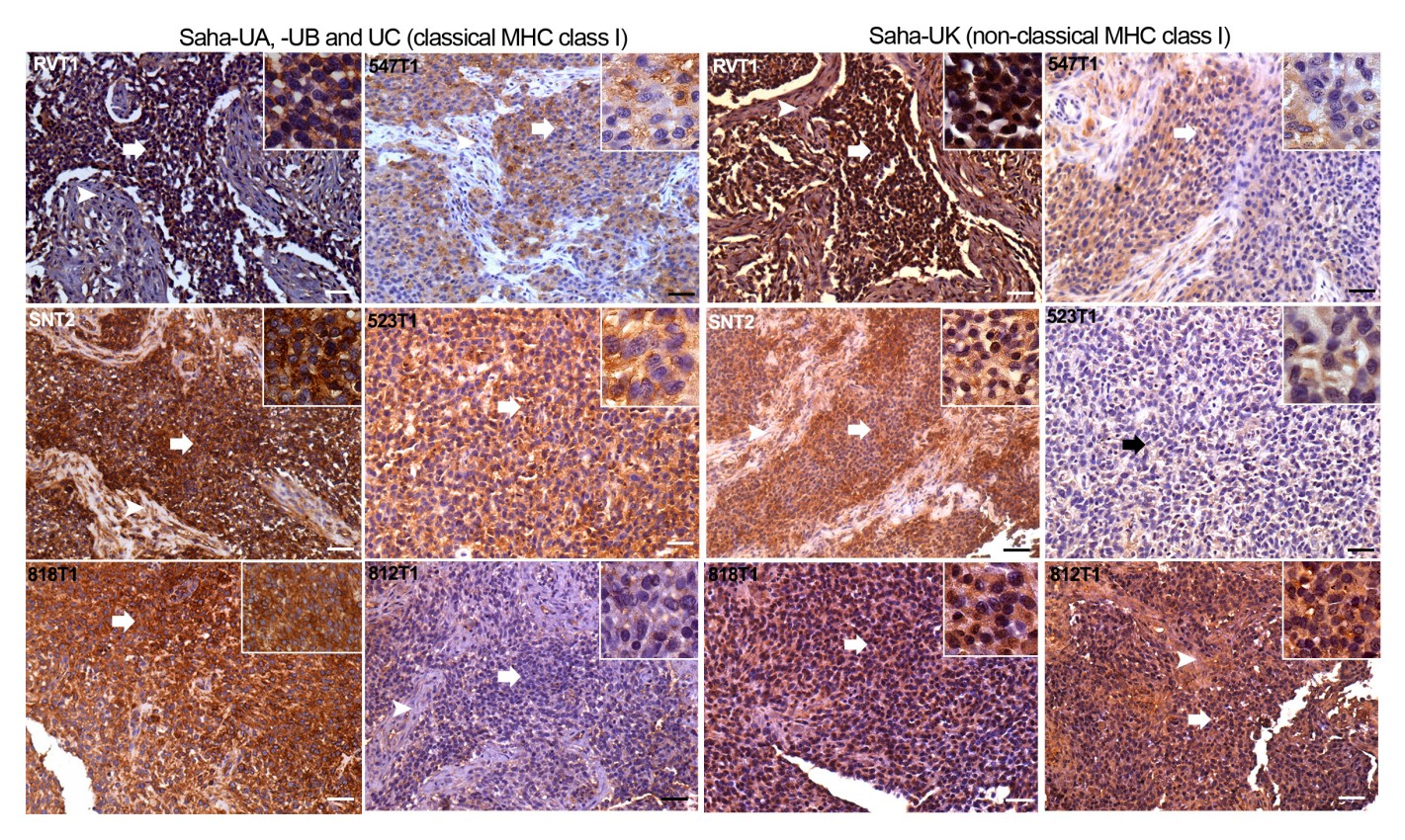

**Figure 3.** DFT2 tumours express variable levels of classical MHC class I (Saha-UA, -UB and -UC) and non-classical MHC class I (Saha-UK) in vivo. IHC staining of DFT2 tumours (DFT2_RVT1, DFT2_SNT2, DFT2_818T1, DFT2_547T1, DFT2_523 DFT2_812) with α-UA/UB/UC_15-25-18 against Saha-UA, -UB and -UC and α-UK_15-29-1 against Saha-UK. Arrows indicate tumour cells for each biopsy; arrow heads indicate stroma separating nests of tumour cells where present. Isotype and secondary antibody controls can be found in *Figure 3—figure supplement 2*. Boxed insets are at 600 x magnification and are taken from areas indicated by arrows. Scale bars represent 50 μm. Positive cells for each marker are stained brown, nuclei are stained blue.

DOI: https://doi.org/10.7554/eLife.35314.006

The following source data and figure supplements are available for figure 3:

**Source data 1.** Nucleotide sequences for the pET22B + SahaUC and pET22B + SahaUK constructs used to test the specificity of the MHC class I antibodies (a- UA/UB/UC 15-25-18 and a- UK 15-29-1) described in *Figure 3*.

DOI: https://doi.org/10.7554/eLife.35314.010

**Figure supplement 1.** Western blots illustrating specificity of the MHC class I heavy-chain monoclonal antibodies.

DOI: https://doi.org/10.7554/eLife.35314.007

**Figure supplement 2.** Isotype controls (IgG1b – anti-SahaUA/UB/UC and IgG2b – anti-SahaUK) and secondary antibody only controls for the immunohistochemistry presented in *Figure 3*.

DOI: https://doi.org/10.7554/eLife.35314.008

**Figure supplement 3.** DFT2 tumours have different growth patterns.

DOI: https://doi.org/10.7554/eLife.35314.009

## DFT2 tumours can have infiltrating CD3 positive cells

The expression of MHC molecules by DFT2 cells may mean that this tumour is more immunogenic than DFT1. As infiltration of lymphocytes has been correlated with a positive prognostic outcome for some cancers (*Galon et al., 2006*; *Chee et al., 2017*) we stained DFT2 tumours for the lymphocyte marker, CD3. Staining of the primary tumours shows that lymphocyte infiltration does occur in some individuals (*Figure 4*). No CD3 positive cells were observed in DFT2_RV or DFT2_547 (*Figure 4*), but a more significant number of CD3 positive cells is evident in DFT2_818, DFT2_523 and DFT2_812 (*Figure 4*), where the lymphocytes are clustered at the stroma and some cells can be seen infiltrating the tumour mass. A small number of CD3 positive cells are visible in DFT2_SN (*Figure 4*).

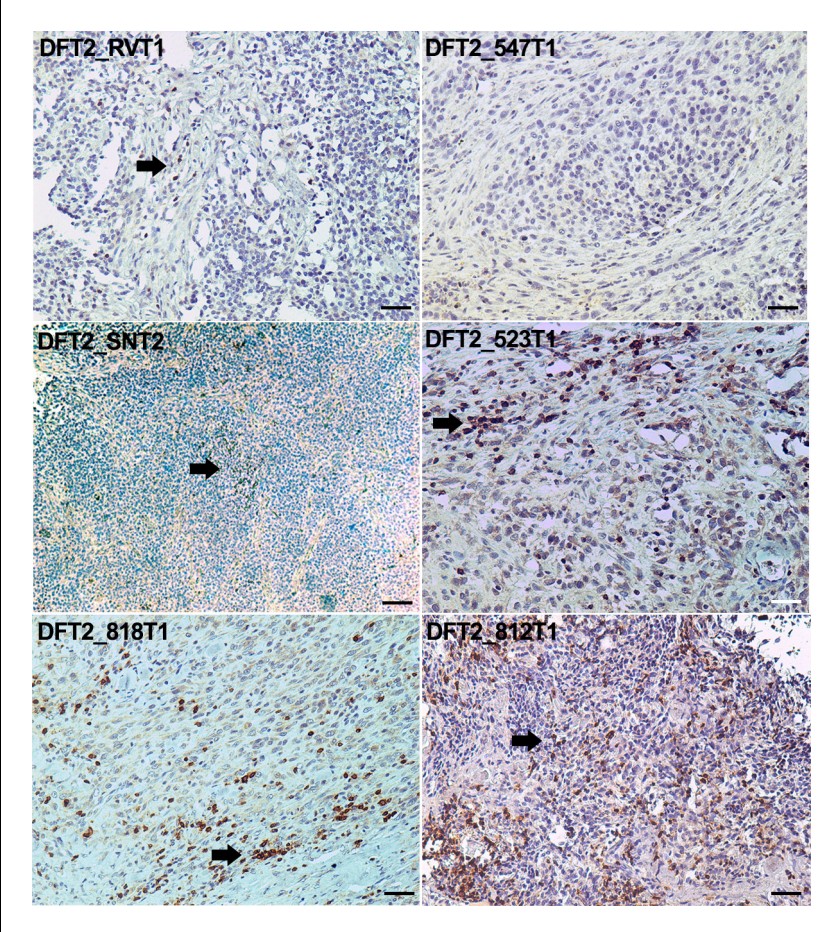

**Figure 4.** CD3 staining of DFT2 tumours. CD3 staining of DFT2_RVT1, DFT2_SNT2 DFT2_818T1, DFT2_TD547T1, DFT2_523T2 and DFT2_812 tumours. CD3 positive cells are indicated by arrows. DFT2_SNT2, and DFT2_547 are CD3 negative. Scale bars represent 50 μm. Positive cells for each marker are stained brown, nuclei are stained blue.

DOI: https://doi.org/10.7554/eLife.35314.011

## DFT2 and DFT1 cells share MHC class I alleles

The classical MHC class I alleles expressed by DFT2 cells should play a role in any host immune response against the tumour. To assess the MHC class I alleles expressed by DFT2 cells we amplified, cloned and sequenced exon 2 (the region of the devil MHC class I where the majority of variation is located) of MHC class I transcripts from DFT2 cell lines. Previous analysis of DFT2 DNA has identified four MHC class I sequences from *Saha-UA*, *Saha-UB* and *Saha-UC* and one sequence from *Saha-UD* present in the DFT2 genome (*Pye et al., 2016b*). We find that DFT2 cells express all five genomic MHC class I sequences, SahaI*35 (*Saha-UA*), SahaI*90 (*Saha-UB*) SahaI*74/88 (unassigned), SahaI*27 (*Saha-UC*) and SahaI*32 (*Saha-UD*) loci (*Table 1* and *Figure 5*). We also identified a sixth expressed sequence (SahaI*27–1) that differs from SahaI*27 by one non-synonymous substitution. Alignment of MHC class I sequences from DFT2 and DFT1 cells (when treated with IFNγ) show that across exon 2 of the heavy chain DFT1 and DFT2 share expression of four MHC class I alleles, SahaI*27, SahaI*90, SahaI*35 and SahaI*27–1, with only SahaI*46 unique to DFT1 and SahaI*74/88 unique to DFT2 (*Table 1*). Using gene specific primers we were able to confirm that both DFT2 and DFT1 cells also express *Saha-UK* (non-polymorphic) (*Table 1* and *Figure 2*).

To assess the relative expression level of each of the classical MHC class I alleles in DFT2 cells, PCR amplicons were cloned and >60 clones were sequenced, resulting in 61 classical MHC class I sequences from DFT2. Of these sequences the most abundant is SahaI*27, representing 22 (36%) of

**Table 1.** MHC class I alleles expressed by DFT2 cell lines and host animals.

The classical and non-classical MHC class I alleles expressed by DFT1, DFT2, TD_RV, TD_818 and TD_SN. Grey boxes indicate the alleles identified in each sample. Numbers in the boxes indicate the number of clones identified for each allele. Sahal*32(UD) and Sahal*UK were amplified with gene specific primers.

| NCBI allele name | DFT1 | DFT2 | TD_RV | TD_818 | TD_SN |
|---|---|---|---|---|---|
| Sahal*46 | | | | | |
| Sahal*27 | | 22 | 22 | 14 | 2 |
| Sahal*27–1 | | 13 | | 25 | 13 |
| Sahal*74/88 | | 9 | | | |
| Sahal*35 | | 13 | | 10 | |
| Sahal*90 | | 4 | | | 1 |
| Sahal*49/82 | | | | 7 | |
| Sahal*97 | | | 6 | | 2 |
| Sahal*33 | | | 6 | | 2 |
| Sahal*36 | | | 2 | | |
| Sahal*37 | | | 5 | | 1 |
| Sahal*29 | | | | 2 | 2 |
| Sahal*32(UD) | | | | | |
| Sahal*UK | | | | | |

DOI: https://doi.org/10.7554/eLife.35314.014

The following source data is available for Table 1:

**Source data 1.** MHC class I transcripts expressed in DFT2 and host devils that were used to generate *Table 1* and *Figure 5*.

DOI: https://doi.org/10.7554/eLife.35314.015

---

the clones sequenced (*Table 1*). Sahal*27–1 and Sahal*35 were represented by 13 clones each (21%), while Sahal*74/88 and Sahal*90 were the least abundant, with 9 (15%) and 4 (6%) clones respectively. These results indicate that DFT2 cells do not express MHC class I alleles equally and that the dominant allele is Sahal*27.

## DFT2 cells express high levels of a MHC class I allele common to its hosts

Given the dominant expression of Sahal*27 by DFT2 cells we determined the prevalence of Sahal*27 and other MHC class I alleles in three host devils. Exon 2 of MHC class I transcripts were amplified from the mRNA of spleen samples from three devils infected with DFT2 (TD_RV, TD_818 and TD_SN) (*Table 1*). Alignment of the sequences demonstrates that the host devils share Sahal*32 (*Saha-UD*), *Saha-UK* and Sahal*27 with DFT2. In addition, TD_818 and TD_SN share Sahal*27–1 and either Sahal*35 or Sahal*90 with DFT2 (*Table 1* and *Figure 5*). The MHC class I alleles that are expressed by DFT2 cells but not by the hosts are of particular interest as they are likely to be immunogenic upon cell transmission. Sahal*74/88 is unique to the DFT2 cell lines and differs from Sahal*27 and Sahal*27–1 found in the host animals by a single non-synonymous substitution at position 59 or position 76, which are not predicted to interact with the TCR (*Figure 5B*). The class I alleles with lower expression on DFT2, Sahal*90 and Sahal*35, share between 91 and 97% amino acid identity with the host alleles.

To identify any further polymorphic sites that may have been missed when selecting clones for sequencing and to validate the sequences found in the hosts we used two approaches. First, we amplified and directly sequenced the MHC class I products from TD_RV, TD_818 and TD_SN to compare the chromatograms produced from sequencing pooled transcripts. This analysis did not reveal any additional polymorphic sites (*Figure 5—figure supplement 1*). Second, we used the

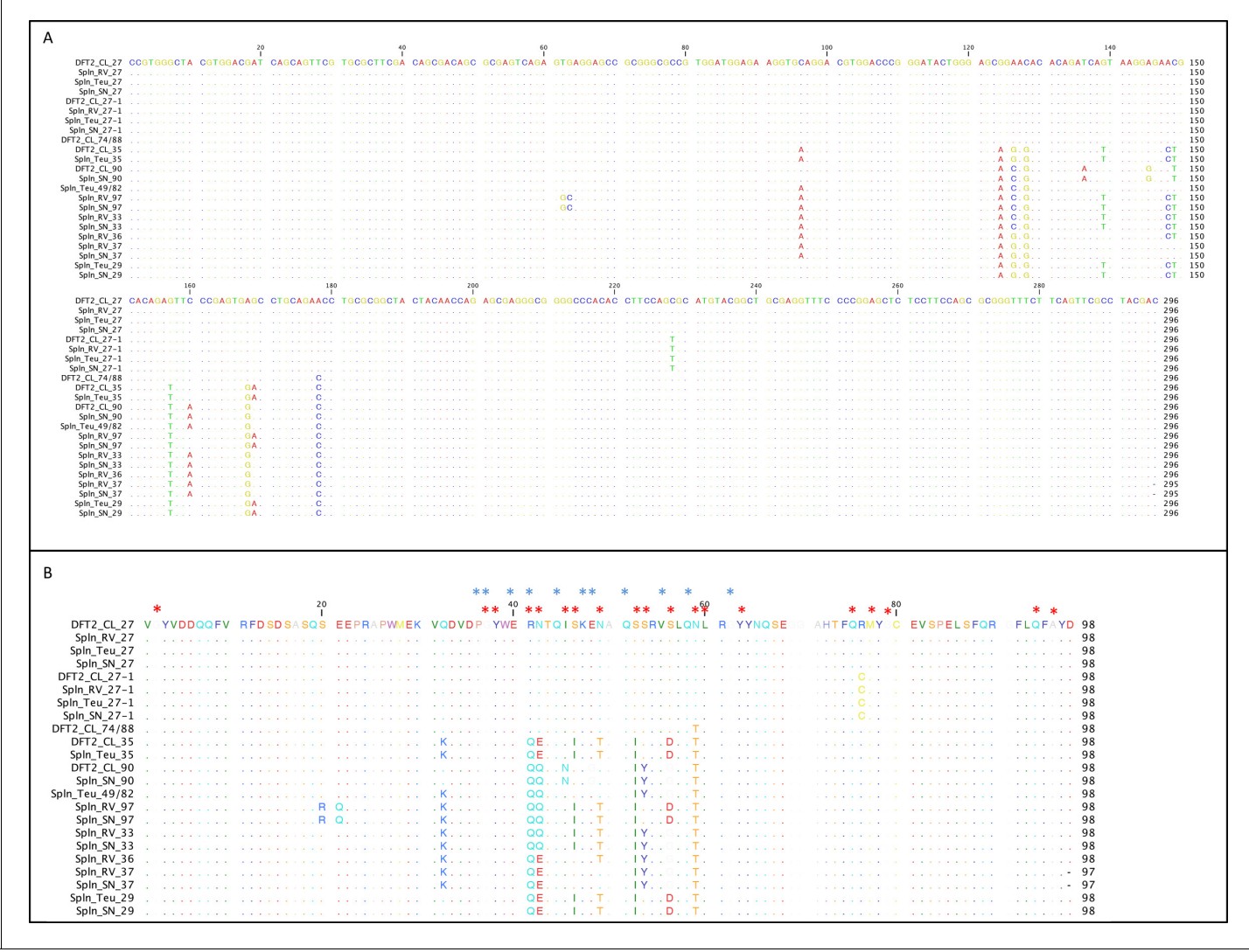

**Figure 5.** DFT2 shares classical MHC class I alleles with its hosts. Alignment of the MHC class I sequences cloned from the mRNA from DFT2 cell lines (DFT2_CL) and host devils (TD_RV, TD_SN and TD_818 (Teu)). Nucleotide alignment in (**A**) and amino acid alignment in (**B**). Red asterisks indicate residues postulated to interact with peptides and blue asterisks indicate residues predicted to interact with TCRs (*Bjorkman et al., 1987*).

DOI: https://doi.org/10.7554/eLife.35314.012

The following figure supplement is available for figure 5:

**Figure supplement 1.** Chromatograms of α2-domian of MHC class I from spleen samples from RV, SN and 818/Teu.

DOI: https://doi.org/10.7554/eLife.35314.013

sequences identified in *Table 1* to conduct targeted searches of the genomes of TD_RV, TD_818 and TD_SN (*Stammnitz et al., 2018*) to confirm the presence of these alleles in genomic DNA.

## Discussion

Here we show that DFT2 cells express classical and non-classical MHC class I molecules both in vitro and in vivo. This is in stark contrast to DFT1, a likely older and more widespread contagious cancer in the Tasmanian devil, which has lost expression of MHC class I (*Siddle et al., 2013*). However, DFT2 cells express the non-polymorphic, non-classical MHC class I molecule (Saha-UK) likely reducing the immunogenicity of these tumour cells. Further, the classical MHC class I allele with the highest expression in DFT2 cells is shared with host devils. Finally, we show that MHC class I expression

varies among DFT2 tumours in vivo, suggesting that MHC class I negative subclones could emerge as DFT2 transmits more widely.

Our results demonstrate that DFT2 cells express MHC class I molecules, in contrast to DFT1 and the contagious cancer circulating among dogs, CTVT. DFT1 cells lack surface expression of MHC class I due to loss of $\beta_2$m, TAP1 and TAP2 transcripts (*Siddle et al., 2013*). However, loss of MHC class I in DFT1 is reversible upon treatment with the inflammatory cytokine IFNγ (*Siddle et al., 2013*) and a small number of individuals have been found to have a successful immune response against the tumour (*Pye et al., 2016a*). Similarly, CTVT is thought to down-regulate MHC expression during transmission, but expression can be upregulated on CTVT cells during a tumour specific immune response from the host (*Yang et al., 1987*). In contrast, we show that DFT2 cells express cell surface $\beta_2$m molecules in addition to classical and non-classical class I heavy chains, indicating that functional MHC class I are present on the surface of DFT2 cells.

The expression of a non-classical MHC class I molecule, Saha-UK, on DFT2 cells could reduce the immunogenicity of the tumour cells. Saha-UK is expressed in some DFT2 tumours in vivo and the *Saha-UK* heavy chain is expressed at similar levels to the three classical MHC class I genes (amplified as a group) in vitro. While antibodies specific for *Saha-UD* are not available, RT-PCR indicates it is also expressed, but at a very low level compared to *Saha-UK*. Non-classical MHC class I molecules can be inhibitory ligands for NK receptors and some human cancers down-regulate classical MHC class I and overexpress non-classical MHC class I to avoid a cytotoxic immune response (*Kochan et al., 2013*). While the function of *Saha-UK* and *Saha-UD* are not known (*Siddle et al., 2009*), neither are highly polymorphic (*Saha-UK* is monomorphic [*Siddle et al., 2010*]) and unlikely to elicit an allogeneic T cell response in host devils. However, further investigation is needed to determine the function of these molecules and the functional classification of Tasmanian devil MHC class I molecules as 'classical' or 'non-classical'.

DFT2 cells also express classical MHC class I alleles belonging to the *Saha-UA*, *-UB* and *-UC* loci. Of these classical MHC class I alleles, SahaI*27 is the most abundant in DFT2 cells and is also found in the three host devils, reducing the immunogenicity of DFT2 cells in these animals. The primers designed in this study should amplify all known DFT1 and DFT2 MHC class I alleles with equal efficiency, suggesting that this allele represents an MHC class I locus with dominant expression. As well as being found in the host devils, this allele is highly prevalent in at least one devil population (*Lane et al., 2012*) and the high expression levels in DFT2 may be facilitating tolerance of the tumour in this population of devils. The similarity of the MHC class I alleles between DFT2 and host devils may reflect a shared geographical range and genotype pool to the cancer founder's (*Stammnitz et al., 2018*). Thus, the frequency of this allele among the wider population may impact the ability of DFT2 to spread further.

The heterogenic expression of MHC class I molecules in DFT2 biopsies suggests that MHC class I expression is not fixed and may be gradually lost in DFT2. While some MHC class I alleles are shared between DFT2 and its hosts, the tumour does have a number of unique alleles. SahaI*74/88 differs to SahaI*27 by only one amino acid that is not predicted to interact with the TCR, but SahaI*90 and SahaI*35 have a number of unique sites when compared to host MHC class I alleles. While these alleles have lower expression on DFT2 than SahaI*27, their presence may still initiate an immune response in some host animals, providing a selective pressure for MHC class I loss. It is possible that, like single organism tumours, DFT2 is undergoing immunoediting (*Dunn et al., 2002*), which can operate to down-regulate specific MHC class I alleles that are immunogenic (*McGranahan et al., 2017*) or select for mutations that remove or modulate total MHC class I expression. However, while four of the six primary DFT2 tumours examined have CD3 positive cells in the tumour mass, which may indicate an immune response, the infiltration is notably present in DFT2_812, which is negative for classical MHC class I and could be expected to be CD3 negative. Thus, further detailed analysis is needed to define the host immune response to DFT2 and its role in shaping tumour evolution.

While DFT1 cells currently circulating in the population are MHC class I negative, it is possible that DFT1 down regulated MHC class I expression after the tumour became transmissible and that initial transmission events were among individuals that shared prevalent MHC class I alleles. This is supported by previous studies that have found reduced polymorphism in devil MHC and microsatellite loci (*Jones et al., 2007*; *Siddle et al., 2007*). In addition, a hemizygous deletion of $\beta_2$m has recently been identified in a DFT1 cell line, perhaps indicating past selective pressure for MHC class I loss in this older transmissible tumour (*Stammnitz et al., 2018*). Similarly, CTVT is thought to have

emerged in an old world dog and the genetic structure of a dog pack would have favoured the emergence of CTVT (*Murgia et al., 2006*). However, the presence or absence of MHC antigens is likely not singly responsible for determining the ability of the host to respond to the tumour. DFT2 may also be manipulating its microenvironment by release of immune suppressive cytokines, such as Transforming Growth Factorβ, which can drive metastasis (*Tauriello et al., 2018*) and in conjunction with PD1/PD-L1 interactions can inhibit and exclude T cells from a tumour (*Mariathasan et al., 2018*). In addition, recruitment of T regulatory cells (*Curiel et al., 2004*; *James et al., 2010*) or immunosuppressive myeloid cells (*Almand et al., 2001*) can also negatively impact the T cell responses to tumours.

DFT1 is now at least 21 years old and CTVT is predicted to be over 10,000 years old (*Murchison et al., 2014*), whereas DFT2 was first discovered in 2014 and is most likely a recently emerged tumour (*Pye et al., 2016b*). As such, DFT2 provides an opportunity to study the early evolution of a contagious cancer. The emergence of a contagious cancer that can transmit while maintaining MHC class I expression indicates that loss of MHC is not necessary for transmission, but as the tumour encounters the immune system of genetically disparate hosts subclones that have downregulated MHC may be selected. Our results predict that loss of MHC class I is already occurring, perhaps due to structural mutations or epigenetic changes. Loss of MHC antigens could allow rapid dissemination of DFT2 through the population, impacting an already vulnerable species.

# Materials and methods

**Key resources table**

| Reagent type (species) or resource | Designation | Source or reference | Identifiers | Additional information |
| --- | --- | --- | --- | --- |
| Gene (*Sarcophilus harrisii*) | Saha-UK | *Murchison et al. (2012)* | ensembl: ENSSHAG00000002942 | Devil_ref v7.0 |
| Gene (*S. harrisii*) | Saha-UC | *Murchison et al. (2012)* | ensembl: ENSSHAG00000000117 | Devil_ref v7.0 |
| Gene (*S. harrisii*) | Saha-UD | *Murchison et al. (2012)* | ensembl: ENSSHAG00000010776 | Devil_ref v7.0 |
| Cell line (*S. harrisii*) | DFT1_4906 | *Siddle et al. (2013)* and *Murchison et al. (2012)* | RRID:CVCL_LB78; DFTD 4906; 86T | Devil Facial Tumour 1; *Supplementary file 1* |
| Cell line (*S. harrisii*) | DFT2_RV | *Pye et al. (2016b)* | DFT2_202T1 | Devil Facial Tumour 2; *Supplementary file 1* |
| Cell line (*S. harrisii*) | DFT2_SN | *Pye et al. (2016b)* | DFT2_203T3 | Devil Facial Tumour 2; *Supplementary file 1* |
| Cell line (*S. harrisii*) | DFT2_549 | This paper | | Devil Facial Tumour 2; *Supplementary file 1* |
| Cell line (*S. harrisii*) (Female) | Fibroblasts_Salem | *Murchison et al. (2012)* | 91 H | Tasmanian devil fibroblasts |
| Cell line (*Cricetulus griseus*) | CHO_SahaIFNy | *Siddle et al. (2013)* | | Chinese Hamster Ovary (CHO) cell line transfected with pcDNA3_SahaIFNy |
| Biological sample (*S. harrisii*) | DFT2_RV | *Pye et al. (2016b)* | DFT2_202T1 | Devil Facial Tumour 2; *Supplementary file 1* |
| Biological sample (*S. harrisii*) | DFT2_SN | *Pye et al. (2016b)* | DFT2_203T2 | Devil Facial Tumour 2; *Supplementary file 1* |
| Biological sample (*S. harrisii*) | DFT2_818 | *Stammnitz et al. (2018)* | | Devil Facial Tumour 2; *Supplementary file 1* |
| Biological sample (*S. harrisii*) | DFT2_547 | *Stammnitz et al. (2018)* | 807T1 | Devil Facial Tumour 2; *Supplementary file 1* |
| Biological sample (*S. harrisii*) | DFT2_523 | *Stammnitz et al. (2018)* | 638T1 | Devil Facial Tumour 2; *Supplementary file 1* |
| Biological sample (*S. harrisii*) | DFT2_812 | *Stammnitz et al. (2018)* | | Devil Facial Tumour 2; *Supplementary file 1* |

*Continued on next page*

*Continued*

| Reagent type (species) or resource | Designation | Source or reference | Identifiers | Additional information |
|---|---|---|---|---|
| Biological sample (*S. harrisii*) | TD_RV | *Stammnitz et al. (2018)* | 202H1 | Tasmanian devil spleen biopsy |
| Biological sample (*S. harrisii*) | TD_SN | *Stammnitz et al. (2018)* | 203 H | Tasmanian devil kidney biopsy |
| Biological sample (*S. harrisii*) | TD_818 | *Stammnitz et al. (2018)* | 818 | Tasmanian deil spleen biopsy |
| Antibody | α-UA/UB/UC_15-25-18; Classical MHC class I Saha-UA, -UB and -UC | This paper | UA/UB/UC_15-25-18 | Antibody recognising MHC class I molecules,SahaUA, UB and UC. Generated using a peptide immunogen (WMEKVQDVDPGYWE). Supernatant from hybridoma used neat. |
| Antibody | α-UK_15-29-1; Non-classical MHC class I Saha-UK | This paper | α-UK_15-29-1 | Antibody recognising MHC class I molecule,Saha-UK. Generated using a peptide immunogen (RITHRTHPDGKVTL). Supernatant from hybridoma used neat. |
| Antibody | IgG1 Isotype control | Sigma Aldrich | clone: MOPC-21; cat number: M5284 | 0.5 mg/ml |
| Antibody | IgG2b Isotype control | Sigma Aldrich | clone: MOPC-141; cat number: M5534 | 0.5 mg/ml |
| Antibody | a-B2m | *Siddle et al. (2013)* | SahaB2m-13-34-48 | supernatant used neat; B2-microglogulin |
| Antibody | a-CD3 | Dako/Agilent | cat number: A0452 | 1:50 |
| Recombinant DNA reagent | pET22B⁺-SahaUC | This paper | | SahaI*UC (SahaI*27) amplified using primer Saha349 and Saha350 (*Supplementary file 2*). |
| Recombinant DNA reagent | pET22B⁺-SahaUK | This paper | | Saha-UK in Pet22B + using primer Saha335 and Saha351 (*Supplementary file 2*). |

## Animals

Six wild Tasmanian devils were either trapped or found dead from road trauma or other causes. Tissue biopsies and fine needle aspirates were either collected post mortem or from live devils that were subsequently released. All animal procedures were performed under a Standard Operating Procedure approved by the General Manager, Natural and Cultural Heritage Division, Tasmanian Government Department of Primary Industries, Parks, Water and the Environment or under University of Tasmania Animal Ethics Committee Permit A0014976. Sample information is shown in *Supplementary file 1*. Tumour and spleen samples were collected as described in *Supplementary file 1* and were formalin fixed (10% neutral buffered formalin) and then paraffin embedded.

## Cell culture conditions and IFNγ treatment

The DFT1 cell line, DFT1_4906, and fibroblast cell line, Fibroblasts_Salem, have been described elsewhere (*Murchison et al., 2012*). Three cell lines derived from DFT2 primary tumours (DFT2_RV, DFT2_SN and DFT2_TD549) were established from fine needle aspirates collected in culture medium and cultured at 35°C and 5% $CO_2$. Cell lines DFT1_4906, DFT2_RV and DFT2_SN were grown in RPMI 1640 with L-glutamine (Gibco; ThermoFisher Scientific) with 10% heat inactivated foetal bovine serum (FBS) (Gibco; ThermoFisher Scientific) and penicillin/streptomycin (100 units/ml penicillin and 0.1 mg/ml streptomycin) (Gibco; ThermoFisher Scientific). Fibroblasts_Salem was cultured in DMEM with high glucose and L-glutamine (Gibco; ThermoFisher Scientific) with 10% FBS (Gibco; ThermoFisher Scientific) and penicillin/streptomycin (100 units/ml penicillin and 0.1 mg/ml streptomycin) (Gibco; ThermoFisher Scientific). Cells were passaged at 80–90% confluency at a ratio of 1:3 using

trypsin (0.05%) to detach cells. DFT1 cells were treated with recombinant devil IFNγ derived from a transfected cell line (CHO_SahaIFNy) as previously described (*Siddle et al., 2013*).

## Flow cytometry

Cells were incubated on ice with anti-devil $\beta_2$m antibody supernatant for 30 min, followed by secondary antibody (2 µg/ml goat anti-Mouse IgG (H + L) Alexa Fluor 488 Conjugate; ThermoFisher Scientific) for 30 min. The specificity of the antibody was confirmed by incubating 50 µg/ml recombinant devil $\beta_2$m protein (*Siddle et al., 2013*) with anti-devil $\beta_2$m antibody on ice for 30 min prior to staining. Cells were analysed on BD FACSCalibur and data analysed using FlowJo software.

## Development of anti-Saha-UK and anti-Saha-UA-UB-UC antibodies

MHC class I transcripts were aligned and translated into protein sequences using CLC workbench. Two regions were identified, in the α1 domain and the α3 domain, where the amino acid sequenced of the Saha-UA, -UB and –UC sequences are highly similar to each other, with only one amino acid change, but contain low similarity to Saha-UK and Saha-UD. Using these sequences the following peptides were synthesised for immunisations, WMEKVQDVDPGYWE against Saha-UA, -UB and -UC and RITHRTHPDGKVTL against Saha-UK. Mice were immunised subcutaneously with a mixture of GERBU adjuvant and 50 µg of either WMEKVQDVDPGYWE-C or RITHRTHPDGKVTL-C coupled to diphtheria toxoid via the N-terminal cysteine. Three days later spleen lymphocytes were fused with the SP2 cell line using PEG as fuseogen. Hybridomas were selected based on reactivity in ELISA against both N- and C-terminal coupled peptide and subsequently screened against Tasmanian devil fibroblast cells and verified against recombinant expressed MHC class I by western blot as described below.

## Electrophoresis and western blotting for antibody screening

Cell pellets were lysed on ice for 30 min at $4 \times 10^7$ cells/ml lysis buffer (150 mM NaCl, 100 mM TrisCl, 1 mM $MgCL_2$ and 1% digitonin) and the lysates clarified. 10 µl of cell lysate was added to 15 µl loading buffer (500 µl 2X Lamelli sample buffer, 50 µl β-mercaptoethanol and 450 µl dd-$H_2O$) and heated to $95^{°C}$ 10 min. Samples were run on 12% ProtoGel (National Diagnostics Protogel 30%) using Laemmli buffers and Fisherbrand Vertical Gel Tank. Proteins were transferred to nitrocellulose blotting membrane (Amersham Protran GE Healthcare Life Sciences) in transfer buffer (25 mM Tris, 190 mM glycine and 20% (w/v) methanol) using Mini Trans-Blot Cell (Bio-Rad). The membrane was blocked for 45 min with 150 mM NaCl, 0.1% Tween 20, 4% milk powder, 50 mM TrisCL, pH 8 and incubated with primary antibodies at 4°C overnight, washed with 150 mM NaCl, 0.1% Tween 20, 4% milk powder, 50 mM TrisCL, pH 8, and incubated with secondary antibody (IRDye 680RD Goat anti-Mouse IgG (H + L)) for 30 min at room temperature before washing as above. Membranes were visualised using the Li-cor Odyssey scanner.

## Specificity of classical MHC class I (Saha-UA, -UB and -UC) and non-classical MHC class I (Saha-UK) antibodies

The specificity of the antibodies was determined using full length recombinant devil MHC class I heavy chain proteins for Saha-UC (SahaI*27) and Saha-UK. SahaI*27 and Saha-UK were amplified from devil fibroblast cDNA using primer Saha349 and Saha350 (Saha-UC) and Saha335 and Saha351 (Saha-UK) (*Supplementary file 2*). The subsequent amplicons were cloned into the pET22b$^+$ vector (Novagen) and transformed into Rosetta pLysS cells (Novagen) according to the manufacturer's instructions and the transcripts were sequenced in both directions. Bacterial colonies containing pET22B$^+$-SahaUC and pET22B$^+$-SahaUK were cultured to $OD_{600}$ 0.6 and protein expression was induced with 1 mM Isopropyl β-D-1-thiogalactopyranoside (IPTG). Bacterial cells were pelleted and resuspended in solubilisation buffer (8 M Urea, 50 mM Mes pH 6.5, 0.1 mM EDTA and 1 mM DTT) and the lysates clarified. The total protein in lysates was measured using Bradford Reagent following the manufacturers instructions. 20 µg protein was loaded onto a 12% gel and electrophoresis and blotting were performed as described above. Membranes were incubated with the undiluted supernatant from primary antibodies α-UK_15-29-1 and α-UA/UB/UC_15-25-18 at 4°C overnight.

## Immunohistochemistry

DFT1 and DFT2 primary tumours were fixed in 10% (mass/vol) PBS-buffered formalin solution for 2 to 4 d. Tissues were processed and embedded in paraffin blocks and cut onto coated slides at 4–5 µm thickness. Sections were deparaffinized in xylene and rehydrated through graded alcohol. Antigen retrieval was performed by water bath (95°C) in citrate buffer solution (10 mM citric acid, 0.05% Tween20 pH 6) for 40 min followed by cooling for 15 min. Endogenous peroxidase was blocked by incubation of slides with 0.3% $H_2O_2$ (Sigma Aldrich) and non-specific protein binding was blocked with 10% (mass/vol) goat serum. Sections were incubated with primary antibody (list of antibodies in *Supplementary file 3*) at 4°C overnight. Peroxidase-coupled secondary antibody (Envision kit; Dako) was used to detect primary antibody binding following the manufacturers instructions. Sections were counterstained with haematoxylin (vector hematoxylin nuclear counterstain Gill's Formula) for 4 min, differentiated in 2% (mass/vol) acetic acid and blued in 0.2% (mass/vol) ammoniated water. Sections were dehydrated through graded alcohol to xylene and cover-slipped. Images were captured using the Nikon Eclipse 400 microscope, Retiga 2000R camera and Q-capture pro seven computer software.

## RT-qPCR

The Nucleospin RNA II kit (Macherey-Nagel) was used to extract RNA from cell lines according to the manufacturer's instructions. 1 µg of RNA was reverse transcribed to cDNA using Thermofischer Scientific RevertAid Premium Reverse Transcriptase (200 U) with 1X RT buffer, 0.5 mM dNTPs, 20 pm oligodT primer (Promega) and nuclease free $H_2O$ to a total volume of 20 µl. Primer set one was designed to amplify the classical MHC class I (*Saha-UA, -UB* and *–UC*) by aligning all available MHC class I sequences from the Tasmanian devil and embedding primers in conserved regions. Primer set one amplifies the three loci as a group, due to the high level of sequence similarity between the classical class I genes, gene specific PCRs for these loci were not possible. Primer set two was designed to amplify the non-classical MHC class I (*Saha-UK*). All primers are listed in *Supplementary file 2*. RT-qPCR was carried out for RPL13A, $\beta_2$m, classical MHC class I (*Saha-UA, -UB* and *-UC*) and non-classical MHC class I (*Saha-UK*) on the StepOnePlus Real-Time PCR system (ThermoFisher Scientific) using PrecisionPLUS qPCR MasterMix (Primerdesign) with primers at 5 mM each following the cycling protocol; 95°C 2 min, followed by 40 cycles of 95°C for 15 s and 60°C for 1 min. RPL13A was used as the reference gene and data analysis was performed using the relative standard curve method for comparing gene expression (DFT1_4906 = 1). All cDNA samples were tested in triplicate and controls with no cDNA template were included. DFT2 at four dilutions (250, 50, 10, 2 and 0.4 ng) was used to create a standard curve of amplification for RPL13A with an $R^2$ of 0.999 and 97.14% efficiency. Recombinant DNA at five dilutions (0.2, 0.04, 0.008, 0.00016 and 0.000032) was used to create a standard curve of amplification of MHC class I *Saha-UK* and MHC class I Saha-UA, UB and UC with $R^2$ values of 0.991 and 0.997% and 105.2 and 111.4% efficiency respectively. IFNγ treated DFT1 (4906) cells at five dilutions (250, 50, 10, 2 and 0.4 ng) was used to create a standard curve of amplification for $\beta_2$m with an $R^2$ of 0.994 and 91.4% efficiency.

## RT-PCR and sequencing of MHC class I transcripts in DFT2, DFT1 and host devils

The Nucleospin RNA II kit (Macherey-Nagel) was used to extract RNA from DFT2 cell lines, DFT1 cells after treatment with IFNγ, and host samples (TD_RV, TD_SN and TD_818). The NucleoSpin totalRNA FFPE kit (Macherey-Nagel) was used to extract RNA from formalin fixed tumour tissues according to the manufacturer's instructions. 1 µg of RNA was reverse transcribed to cDNA as described previously. Three primer sets were used to amplify all classical and non-classical MHC class I alleles. Primer set one was used to amplify *Saha-UA, -UB* and *–UC* products as a group, primer sets 2 and 3 were used to amplify *Saha-UK* and *Saha-UD* genes specifically (for PCR conditions see *Supplementary file 4*). All products were purified using NucleoSpin Gel and PCR Clean-up (Macherey-Nagel), cloned into a pJET plasmid (CloneJET; ThermoFisher Scientific). Plasmid DNA from individual clones was isolated and sequenced using T7 primer. PCR, cloning and sequencing was performed in triplicate for the DFT2 cells and duplicate for the spleen samples. For primer set 1 between 15 and 65 clones were sequenced for each PCR, for primer sets 2 and 3 between 4 and 6

clones were sequenced. To assess the relative abundance of each of the classical MHC class I alleles within DFT2 cells 65 clones were isolated and sequenced using primer set 1.

To validate the MHC class I alleles, *Saha-UA*, *-UB* and *-UC* genes were amplified using primer set one as described above and resulting PCR products (containing all MHC class I alleles for these genes) were sequenced directly using these primers. The resulting chromatograms were used to identify polymorphic sites in the transcripts and to confirm the validity of alleles. In addition, MHC class I alleles for RV, 818 and SN were searched against genomes generated for these animals (*Stammnitz et al., 2018*) to validate their presence in the genome. All sequences were quality checked and analysed in CLC Genomic Workbench. Residues in the MHC class I sequences predicted to interact with the peptide or the TCR were identified by aligning to HLA-A2 (*Bjorkman et al., 1987*).

## Acknowledgements

The authors thank the histology department at the University of Southampton Hospital for H and E staining and paraffin embedding some tumour samples. This work was supported by a Research Project Grant from Leverhulme Trust (RPG-2015–103) to HVS and TE. AC is supported by the Gerald Kerkut Charitable Trust and University of Southampton Vice-Chancellor's scholarship. EPM, MRS and YMK are supported by the Wellcome Trust (102942/Z/13/A) and a Philip Leverhulme Prize awarded by the Leverhulme Trust. MRS is supported by a scholarship from the Gates Cambridge Trust and YMK is supported by a Herchel Smith Postgraduate Scholarship.

## Additional information

### Funding

| Funder | Grant reference number | Author |
|---|---|---|
| Gerald Kerkut Charitable Trust | Postgraduate student scholarship | Alison Caldwell |
| University of Southampton | Vice-Chancellor's Scholarship | Alison Caldwell |
| Gates Cambridge Trust | | Maximilian R Stammnitz |
| Wellcome | 102942/Z/13/A | Elizabeth P Murchison Maximilian R Stammnitz Young Mi Kwon |
| Leverhulme Trust | Philip Leverhulme Prize | Elizabeth P Murchison Maximilian R Stammnitz Young Mi Kwon |
| Leverhulme Trust | RPG-2015-103 | Tim Elliott Hannah VT Siddle |
| Herchel Smith Postgraduate Scholarship | | Young Mi Kwon |

The funders had no role in study design, data collection and interpretation, or the decision to submit the work for publication.

### Author contributions

Alison Caldwell, Formal analysis, Investigation, Writing—original draft; Rachel Coleby, Cesar Tovar, Marios Tringides, Investigation; Maximilian R Stammnitz, Resources, Validation, Methodology; Young Mi Kwon, Validation; Rachel S Owen, Investigation, Writing—review and editing; Elizabeth P Murchison, Resources, Validation, Writing—review and editing; Karsten Skjødt, Resources, Supervision, Writing—review and editing; Gareth J Thomas, Formal analysis, Investigation; Jim Kaufman, Gregory M Woods, Resources, Investigation, Writing—review and editing; Tim Elliott, Conceptualization, Supervision, Writing—review and editing; Hannah VT Siddle, Conceptualization, Supervision, Funding acquisition, Investigation, Writing—original draft, Writing—review and editing

### Author ORCIDs

Alison Caldwell http://orcid.org/0000-0003-2116-2482
Rachel Coleby http://orcid.org/0000-0002-2490-0199
Maximilian R Stammnitz http://orcid.org/0000-0002-1704-9199
Elizabeth P Murchison http://orcid.org/0000-0001-7462-8907
Gareth J Thomas http://orcid.org/0000-0003-3832-7335
Gregory M Woods https://orcid.org/0000-0001-8421-7917
Hannah VT Siddle http://orcid.org/0000-0003-2906-4385

### Ethics

Animal experimentation: All animal procedures were performed under a Standard Operating Procedure approved by the General Manager, Natural and Cultural Heritage Division, Tasmanian Government Department of Primary Industries, Parks, Water and the Environment or under University of Tasmania Animal Ethics Committee Permit A0014976.

### Decision letter and Author response

Decision letter https://doi.org/10.7554/eLife.35314.026
Author response https://doi.org/10.7554/eLife.35314.027

## Additional files

### Supplementary files

• Supplementary file 1. Biological samples used in this study. Details of the tumour and host tissue samples, the date the samples were collected from the animal and the location of the animal when it was trapped. *As described previously (*Pye et al., 2016b*).
DOI: https://doi.org/10.7554/eLife.35314.016

• Supplementary file 2. Primers used in this study with amplicon size and optimised annealing temperature.
DOI: https://doi.org/10.7554/eLife.35314.017

• Supplementary file 3. Antibodies used in this study.
DOI: https://doi.org/10.7554/eLife.35314.018

• Supplementary file 4. PCR conditions for the primers used in this study.
DOI: https://doi.org/10.7554/eLife.35314.019

• Transparent reporting form
DOI: https://doi.org/10.7554/eLife.35314.020

### Data availability

All data generated or analysed during this study are included in the manuscript and supporting files. Source data files have been provided for Table 1, Figure 5 and Figure 3-figure supplement 1.

The following previously published datasets were used:

| Author(s) | Year | Dataset title | Dataset URL | Database, license, and accessibility information |
|---|---|---|---|---|
| Murchison | 2012 | Devil_ref v7.0 (GCA_000189315.1) | http://www.ensembl.org/Sarcophilus_harrisii/Info/Index | Publicly available at the European Nucleotide Archive (accession no: GCA_000189315.1) |
| Stammnitz | 2018 | Genomes of Tasmanian devil transmissible cancers DFT1, DFT2 and normal animals | https://www.ebi.ac.uk/ena/data/view/PRJEB21902 | Publicly available at the European Nucleotide Archive (accession no: ENA: PRJEB21902) |

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
