## [Decision Letter]

Thank you for submitting your article "The newly-arisen Devil Facial Tumour disease 2 (DFT2) reveals a mechanism for the emergence of a contagious cancer" for consideration by *eLife*. Your article has been reviewed by Arup Chakraborty as the Senior Editor, a Reviewing Editor, and three reviewers. The following individuals involved in review of your submission have agreed to reveal their identity: Mel Greaves (Reviewer #1); Rob Miller (Reviewer #2); Cornelis Melief (Reviewer #3).

The reviewers have discussed the reviews with one another and the Reviewing Editor has drafted this decision to help you prepare a revised submission.

While all the referees found the subject of interest and the findings of relevance to understanding immune – tumor interactions, they also found limitations to the study in its current state. The major areas of concern involved the immunohistochemical findings as well as some of the conclusions that were reached without sufficient corresponding data.

Essential Revisions:

1) Detailed and specific PCR-based analysis of MHC locus expression in DFT2 cells was carried out on cell lines (#3). In vivo analysis was more limited and based on antibody screening (and revealed variable cell-cell expression). Are the cell lines representative of the in vivo tumor? If MHC down-regulation in vivo is epigenetic, then expression in vitro could be misleading. The paper would be substantially strengthened if it is possible to provide some validation of the cell line data using tissue samples. In this regard, patchiness of the immunohistochemical staining seems odd. Why should there be areas of continued expression and areas where there is loss of expression? Why wouldn't loss of expression be more random? Some of the images seem to show that staining is peripheral in to the tumors (Figure 4—figure supplement 1). Was this the case and why might that be? Because the older sections of the tumor were more likely to have lost expression? Were sequential sections stained with anti-UA, B, C followed by anti-UK to see if staining is common to areas and therefore may be artifact or some evidence of co-staining?

2) It is unclear how the authors think it likely DFT2 succeeds in immune evasion. Is it (a) there is nothing to be recognised on DFT2 cells; or (b) the DFT2 cells are immune-suppressive? The suggestion that non-classical MHC class 1 molecules (*Saha*) are inhibitory appears to be entirely speculative. Can the authors provide further studies that might illuminate the mechanism?

In this regard, MHC molecule expression in vivo is recorded as variable. From this the authors conclude that they may be catching immune-editing or evolutionary selection for MHC loss as it happens. This could be true, but it is unpersuasive and very speculative; there is considerable variation in expression and no evidence for emerging dominance of MHC negative variants. Importantly, the authors do not observe infiltration of tumor with lymphocytes which might reflect ongoing selection. How do they reconcile this with their hypothesis about immune evasion and selection? One would expect processing and presentation in the MHC I molecules of minor histocompatibility antigens, at least some of which are expected to differ between the first animal from whom the tumor arose and subsequent recipients. Therefore, other reasons such as a T cell hostile micro-environment despite the presence of minor H differences may play a role in the selection for transmission of the cancer. This could include a variety of mechanisms also reported for immunogenic cancers in autologous hosts:

1) Exclusion of T cells by TGF**β** (two recent Nature papers)

2) High numbers of immunosuppressive myeloid cells

3) Lack of the proper chemokines and/ or chemokine receptors

4) Hypoxia

5) IDO expression

The authors are encouraged to discuss this and refer to relevant review papers cataloguing these possibilities for autochthonous tumors. If this is researched further, it may lead to therapeutic possibilities that are also relevant for immunotherapy of cancer in general.

---

## [Author Response]

Essential Revisions:

*1) Detailed and specific PCR-based analysis of MHC locus expression in DFT2 cells was carried out on cell lines (#3).* In vivo *analysis was more limited and based on antibody screening (and revealed variable cell-cell expression). Are the cell lines representative of the* in vivo *tumor? If MHC down-regulation* in vivo *is epigenetic, then expression* in vitro *could be misleading. The paper would be substantially strengthened if it were possible to provide some validation of the cell line data using tissue samples.*

This is an important point and we do recognise that comparing cell lines to in vivo tumours can be misleading. To address this we have used reverse transcriptase (RT)-PCR to compare the expression of B2m, classical MHC class I and the non-classical class I in the three DFT2 cell lines and the 6 primary tumour samples which were used in the immunohistochemistry (IHC) experiments (Figure 2E). Two of these primary tumours, DFT2_RV and DFT2_SN were used to derive the cell lines analysed in Figure 1 and are key to comparing the flow cytometry, immunohistochemistry and RTPCR results. This data has been added to Figure 2E and the Results section.

We find that the primary tumour samples express B2m, reflecting the B2m cell surface protein and mRNA expression in the cell lines. In addition, the primary tumours express both classical MHC class I and non-classical MHC class I, but this expression appears weaker than the cell lines and variable between the tumours. While variable expression was also overserved in the IHC experiments, it should be noted that we do not consider the RT-PCR quantitative as the amount of stroma (host) will vary in each sample, contributing to the total RNA and potentially affecting transcript levels. This has been noted in the manuscript (subsection “DFT2 cells express classical and non-classical MHC class I heavy chain genes in vitroand in vivo.”).

In this regard, patchiness of the immunohistochemical staining seems odd. Why should there be areas of continued expression and areas where there is loss of expression? Why wouldn't loss of expression be more random? Some of the images seem to show that staining is peripheral in to the tumors (Figure 4—figure supplement 1). Was this the case and why might that be? Because the older sections of the tumor were more likely to have lost expression?

This point is valid and we have performed a review of our immunohistochemistry data to identify regions subject to edge artefact and necrosis of the tumour tissue, which may have given a misleading impression of positive expression. We have re-imaged the sections, avoiding these areas and shown the six different tumours in Figure 3 (rather than as supplementary material) to present these results more clearly.

Figure 3 shows that DFT2 primary tumours can have high expression of the classical class I heavy chains (i.e. DFT2_RV, SN and 818), while others have lower expression (i.e. DFT2_812). Variable expression is also observed for the non-classical MHC class I, *Saha-UK*. However, the staining indicates that expression may be primarily internal, and this is reflected in the text of the Results section, Figure 3.

In a number of tumours, the staining for the classical MHC class I and non-classical MHC class I (i.e. DFT2_547) is ‘patchy’ we see some areas of the tumour tissue where cells are staining more strongly for the MHC class I proteins. This may be because older sections of the tumour have lost expression, but it may also be due to local environmental factors, signalling and cytokines from host devils. However, further experiments are needed to assess the mechanisms controlling modulation.

There is considerable precedent for this in single organism tumours and we have included for the reviewers some examples of head and neck tumours with a range MHC class I expression, including weak, variable and strong examples (stained with W632, a pan specific HLA antibody). There is also considerable literature on loss of MHC class I molecules through DNA mutations in genes involved in antigen processing and presentation and so called ‘soft’ mutations, usually epigenetic and sometimes rescueable (Restifo et al., 1996; McGranahan et al., 2017). We have referred to this literature in the Discussion section.

Were sequential sections stained with anti-UA, B, C followed by anti-UK to see if staining is common to areas and therefore may be artifact or some evidence of co-staining?

Where possible we have always used serial sections for the staining of class I heavy chains and CD3 (Figure 3 and Figure 4). However, in some cases the tissue architecture is different between sections, particularly in tumours with little fibrous stroma. We have attempted co-staining for the different heavy chains, but as both the classical and non-classical genes are expressed together on cells and the affinity of the antibodies is different, it is difficult to resolve both signals clearly.

2) It is unclear how the authors think it likely DFT2 succeeds in immune evasion. Is it (a) there is nothing to be recognised on DFT2 cells; or (b) the DFT2 cells are immune-suppressive? The suggestion that non-classical MHC class 1 molecules (Saha) are inhibitory appears to be entirely speculative. Can the authors provide further studies that might illuminate the mechanism?

A combination of both a and b. As the reviewers point out below there are many ways in which tumours can actively suppress the immune system, but in this work we have focused on what would be antigenic on the cell surface in the context of MHC class I. Unfortunately, at present we cannot contribute any other data to determine the function of the non-classical MHC class I, this will have to remain speculative. However, we have pointed out in the Discussion section that its non-polymorphic nature means it is unlikely to be a major alloantigen in devils, regardless of function.

*In this regard, MHC molecule expression* in vivo *is recorded as variable. From this the authors conclude that they may be catching immune-editing or evolutionary selection for MHC loss as it happens. This could be true, but it is unpersuasive and very speculative; there is considerable variation in expression and no evidence for emerging dominance of MHC negative variants. Importantly, the authors do not observe infiltration of tumor with lymphocytes which might reflect ongoing selection. How do they reconcile this with their hypothesis about immune evasion and selection?*

We agree that we have not shown extensive data on how the devil immune system responds to DFT2 (if it does at all). We have improved this analysis by undertaking more extensive staining for CD3 to screen for infiltrating lymphocytes in the primary tumours (This is now shown in Figure 4). We find that 4 of the 6 primary tumours have CD3 cell infiltrating the tumour (Results section).

2

Previous reports have shown the vast majority of DFT1 tumours are CD3 negative (reflecting the lack of immune response to this tumour) and as we have noted in the text lymphocyte infiltration is associated with a more positive prognostic outcome in many single organism tumours (Chee et al., 2017 and Galon et al., 2006). At present the proposal that DFT2 is losing MHC expression due to an immune response from the host is speculative and we have altered the Discussion section for clarity.

One would expect processing and presentation in the MHC I molecules of minor histocompatibility antigens, at least some of which are expected to differ between the first animal from whom the tumor arose and subsequent recipients. Therefore, other reasons such as a T cell hostile micro-environment despite the presence of minor H differences may play a role in the selection for transmission of the cancer. This could include a variety of mechanisms also reported for immunogenic cancers in autologous hosts:*1) Exclusion of T cells by TGF***β** (two recent Nature papers)2) High numbers of immunosuppressive myeloid cells3) Lack of the proper chemokines and/ or chemokine receptors4) Hypoxia5) IDO expressionThe authors are encouraged to discuss this and refer to relevant review papers cataloguing these possibilities for autochthonous tumors. If this is researched further, it may lead to therapeutic possibilities that are also relevant for immunotherapy of cancer in general.

This is a good point and while our current studies have focused on MHC expression in DFT1 and DFT2 this is an area of importance. We have included reference to some of these mechanisms in the Discussion section.